# Sustainable palm oil certification inadvertently affects production efficiency in Malaysia
Nina Zachlod [1] ✉, Michael Hudecheck [1], Charlotta Sirén [1] & Gerard George [2,3]

Sustainability certifications have rapidly gained prominence and become standards across many industries, yet knowledge about the potential unintended consequences of their criteria remains limited. Here, we use European Space Agency multispectral imagery satellite data in combination with economic and location data to investigate whether the certification process for palm oil production results in unintended consequences. Our results indicate decreases in plantation efficiency both prior to and following the certification obtainment. Our findings highlight the importance of considering possible unintended consequences of sustainability certifications beyond their immediate goals and criteria.

Sustainability certifications have rapidly gained prominence for governing both social and environmental aspects of trade in the private sector[1]. They have become industry standards[2], yet very little is known about their potential unintended consequences on sustainability and about the environmental impact of their criteria[3]. Sustainable palm oil production is crucial, since unsustainable palm oil production and cultivation practices have led to deforestation, reduced biodiversity[4], and adverse impacts on endangered species' habitats, and they have exacerbated the effects of global warming and climate change[5]. Voluntary certification schemes, such as the Roundtable on Sustainable Palm Oil (RSPO) certification, aim for producer compliance with sustainability criteria and thereby serve as market-based instruments of sustainability governance[6]. The RSPO is a non-profit, globally operating organization that has developed and issued environmental and social criteria for company compliance, intended to minimize negative impacts of palm oil production on the local environment, wildlife, and communities[7]. These criteria take the form of the 2018 RSPO Principles and Criteria (RSPO P&C 2018) and the 2019 RSPO Independent Smallholder Standard, with the final draft documents for the Standards Revision 2022–2024 available on the RSPO website as of January 2025[8]. The RSPO comprised 6106 active members in 105 countries and territories and 505 certified palm oil mills under 97 certified growers, amounting to a total certified area of 5,151,975 ha[9] on January 8, 2025, making it one of the most widely used certificates in the palm oil industry.

In this study, we concentrate on the palm oil industry in Malaysia, as it was among the 15 countries with the largest average forest loss between 2005 and 2013, with 69% of deforestation attributable to palm oil[10]. Malaysia is the second-largest producer of palm oil worldwide, supplying around one third of global palm oil[11]. In 2024, Malaysia has a total mature certified sustainable palm oil production area of 920,173 ha (RSPO, 2024), approximately 2.78% of the total land mass[5]. Palm oil is a primary industry in Malaysia and contributes 2.4% to its overall GDP[12]. In 2023/24, Malaysia produced a total of 19,000 metric tons of palm oil. To facilitate sustainable palm oil production, the Malaysian government has strongly encouraged Malaysian oil palm plantations to obtain RSPO certification[13]. Despite its wide acceptance, to the best of our knowledge, there currently is little insight into or evidence on the longitudinal impact of RSPO certificates on oil palm plantation efficiency before and after certification obtainment.

Recent studies have focused predominantly on the socioeconomic[14] and financial[15] impacts of the RSPO. The primary reason for the lack of studies on the environmental consequences of palm oil sustainability certificates and RSPO certification specifically, to our understanding, is the demanding nature of data collection in this study context. The few studies done, to the extent of our knowledge, have concentrated on Indonesia, where the palm oil industry has been associated with issues of legality and self-reporting practices[16]. Previous research on the effect of the RSPO on people and the environment has unveiled heterogeneous effects of the certification on deforestation and forest protection in Indonesia[17]. Other studies have investigated comparable certificates and voluntary certification programs in other agricultural sectors. A study of the Forest Stewardship Council in Kalimantan, Indonesia[18], has found a positive environmental impact of the certification program, reducing aggregate deforestation by five percentage points between 2000 and 2008. Whilst it had no statistically significant impact on e.g., fire incidences, it reduced the incidences of air pollution by 31% in the observed time period. A study on the reduced emissions from deforestation and forest degradation programs across the Global South[19] has found positively significant yet moderately sized average

[1]Institute of Responsible Innovation, University of St.Gallen, St. Gallen, Switzerland. [2]Georgetown University, Washington, DC, USA. [3]International Medical University, Kuala Lumpur, Malaysia. ✉e-mail: nina.zachlod@unisg.ch

environmental impacts and welfare-neutral to slightly positive socio-economic impacts. Research on other (non-RSPO) natural resource policies designed to curtail production's environmental impact however found that they may result in unintended consequences, such as the displacement of environmental impacts[20].

Tracking the implementation of and adherence to the criteria of the RSPO has low levels of reliability due to insufficient governance inside plantation sites[21]. The reliance on third parties for auditing and verifying compliance with the standards and corresponding data collection also poses challenges in this regard, as auditors tend to dismiss evidence of non-compliance put forward by local communities[22], undermining attempts at data verification. Due to the inherent difficulties in the composition of reliable and comprehensive, large-scale, longitudinal datasets on the actual impact of such certificates, it remains difficult to confirm with traditional data whether sustainability certificates, such as the RSPO certificate, are effective governing mechanisms.

It is thus crucial to (a) investigate the effects of the RSPO on the impact of palm oil production in Malaysia as the second-largest palm oil producer worldwide to broaden previous findings focused on Indonesia as the largest palm oil producer globally, and (b) obtain access to reliable and incorruptible data on the actual environmental impact of the RSPO on palm oil production[23].

In this context, satellite imagery can provide incontrovertible evidence of the production sites and has previously been used to track the expansion of oil palm plantations in Indonesia and Malaysia[24]. Therefore, we combine European Space Agency (ESA) multispectral imagery satellite data with economic and location data sources to investigate whether the RSPO certification process results in unintended consequences due to means-end decoupling[25], i.e., producers being trapped into producing outcomes and responses that are not explicit goals of the certification and are ill suited to addressing the challenges of sustainable production. We specifically ask how preparing for and obtaining RSPO certification affected the efficiency of one of Malaysia's largest palm oil producers both before and after certification.

The findings of our study have several implications for the literature on sustainable production. They amplify previous findings on the need for spatial prioritization for certification expansion and consolidation[26] by unveiling additional requirements for certification criterion re-evaluation. Our results encourage further empirical exploration of market-based, voluntary sustainability governance mechanisms and critical consideration of their consequences both in theory and practice. Issuers of market-based sustainability standards must carefully evaluate all potential impacts and consequences of their criteria beyond those initially intended and examine their certifications for unintended, potentially sustainability-harming consequences both before and after certification obtainment.

Plantation owners involved in palm oil production have strong incentives to obtain sustainability certificates as competitors within the industry obtain them. Zero-deforestation commitments to reduce emissions and protect biodiversity have become common amongst palm oil producers[27]. In this context, the RSPO P&C 2018 are focused on the protection of forests from legal and illegal deforestation through no deforestation, no peat, and no exploitation commitments[25]. Given the popularity of the RSPO and other sustainable production certificates amongst consumers[28], major palm oil producers globally are encouraged to obtain the certification, with 7101 facilities holding RSPO supply chain certificates[9] as of January 8, 2025.

However, it has been shown that sustainable governance policies enacted to protect the environment in various policy contexts and resource sectors could result in a variety of unintended consequences[16]. Alternative, market-based instruments of sustainability governance, such as sustainability standards, certificates, or nature conservation programs, have also been suggested to cause negative externalities, such as hidden costs and unacknowledged impacts in the form of e.g., moving deforestation to properties not belonging to anti-deforestation schemes[29,30]. Although the RSPO P&C 2018 does not mandate a reduction in plantation efficiency (i.e., the number of oil palm trees on the plantation), its role as a global, market-based mechanism for sustainability governance[4] may inadvertently lead to such negative externalities. These externalities may manifest during the certification process through e.g., diminished plantation efficiency due to unintended consequences of their mandated criteria. Once a production site has been certified against a set of sustainability standards, annual re-audit procedures are designed to ensure ongoing compliance. The limiting compliance policies of the certification therefore continue to affect the production site beyond the certification.

Following the strategies of their major competitors and the increasing pressure from consumers, CorpPalm (a pseudonym for the palm oil producer we study) has publicly claimed certification of all their plantations in the Sabah Region in Malaysia is a key component of their sustainability strategy. CorpPalm applied for and obtained certifications for all of their plantations during the seven-year timeframe of our study. It is very likely that notable changes in the production strategy of CorpPalm plantations will have occurred both prior to and after their certification obtainment, as CorpPalm needed to adjust to the certification requirements. Based on previous findings on other types of unintended consequences and negative externalities of market-based instruments of sustainability governance, in this study we expect the certification to have resulted in observable changes in plantation efficiency measured as the number of oil palm trees on the plantations of CorpPalm.

## Results
For the purpose of our study, we analyzed a total of 144 plantations in the Sabah region of Malaysia between 2017 and 2023 (cf. Table 1 and Fig. 1). We subset our geospatial analysis to the predefined spatial regions of the oil palm plantation boundaries. The plantations were grouped under six mills with varying numbers of plantations per mill, which are operated by CorpPalm, of which three were certified under the RSPO in 2018 and three in 2023. Of the plantations, 28 are CorpPalm's own estates, 39 are outside supplier estates and 77 are managed by small growers as outside suppliers. One third of our sample (48 plantations) was certified in 2018 under the P&C 2018 certification criteria, the remaining two thirds (96 plantations) in 2023. For our analysis we split our sample into those plantations that obtained the certification in 2018 and those that obtained it in 2023. This allows us to observe the effect of certification obtainment before (Sample 1) and after certification (Sample 2) issuance. Whilst roughly half of the observed plantations fall under the small-grower scheme, they represent only 12,188 ha of the observed plantation area for our study. This is expected, given the comparatively small size of small-grower plantations in the palm oil industry[31] and emphasizes the comparative importance of corporate decisions affecting plantation management for large own and outside supplier held estates.

## Table 1 | Overview of plantations

| Plantation type | Own estate | Outside supplier estate | Outside supplier small-grower | Total |
|---|---|---|---|---|
| Number of plantations | 28 | 39 | 77 | 144 |
| Certified in 2018 | 5 (17.9%) | 11 (28.2%) | 32 (41.6%) | 48 (33.3%) |
| Certified in 2023 | 23 (82.1%) | 28 (71.8%) | 45 (58.4%) | 96 (66.7%) |
| Total area (ha) | 33,026 | 90,469 | 12,188 | 135,683 |

a

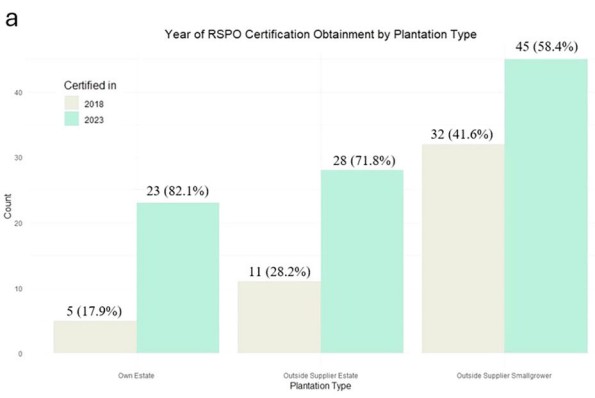

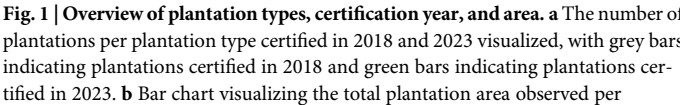

b

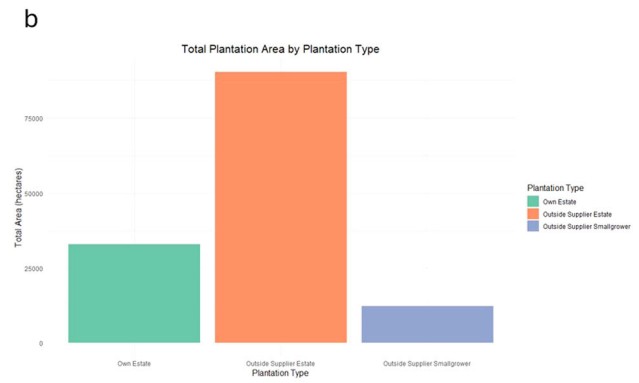

**Fig. 1 | Overview of plantation types, certification year, and area. a** The number of plantations per plantation type certified in 2018 and 2023 visualized, with grey bars indicating plantations certified in 2018 and green bars indicating plantations certified in 2023. **b** Bar chart visualizing the total plantation area observed per plantation type, with own estate represented in green, outside supplier estate represented in orange and outside supplier small-grower plantations represented in purple. These figures were created using the R packages dplyr[56], tidyr[57], and ggplot2[58].

To analyze the possible influence of environmental conditions on the oil palm plantation coverage and to rule out alternative explanations for changes in production efficiency, we introduced several control variables. Table 2 presents the results for the random-effect models relative to the baseline level 2018. Models 1 and 3 in Table 2 report the effects of our control variables on oil palm plantation efficiency. We controlled for the annual trade price for Malaysian palm oil on Dec 31 as the exchange date per year as an alternative explanation pertaining to changes in demand for the palm oil produced on the plantations under observation. The trade price fluctuated over the study period, reaching its minimal value in 2018 at 1940 MYR per metric ton and its maximum value in 2021 at 5,046 MYR per metric ton. Palm oil price has a statistically significant positive coefficient (Sample 1: $\beta = 0.0001$, $p < 0.001$; Sample 2: $\beta = 0.0001$, $p < 0.05$). The positive beta suggesting higher tree coverage is expected, as oil palm trees are unlikely to be cut down when oil prices are high. This is because it is the fruit of the tree, rather than the tree itself, that is used in palm oil production.

Poor weather conditions, such as drought, can lead to lower vegetation coverage. To control for such phenomena, we also calculated and compared three indices commonly employed for monitoring land cover and vegetation development. The indices are calculated based on the bands in our multispectral satellite data. The Normalized Difference Vegetation Index (NDVI) indicates vegetation health, the Normalized Difference Moisture Index (NDMI) indicates soil moisture levels, and the Bare Soil Index (BSI) indicates soil visibility levels. The bare soil index can be used to determine crop status, i.e., whether plants are (already) growing, or to detect deforestation. All values are relatively stable over our study period, showing minor fluctuations only and ranging within the expected values for all indices for comparable agricultural areas (cf. Table 3).

Vegetation health and groundwater levels are stable throughout the study period, indicating no issues with vegetation health or water availability. Vegetation health (Sample 1: $\beta = 0.753$, $p < 0.05$; Sample 2: $\beta = 0.751$, $p > 0.05$) has a significant effect only on the coverage of plantations certified in 2023. Groundwater availability has no statistically significant effect on either sample (Sample 1: $\beta = 2.349$, $p > 0.05$; Sample 2: $\beta = 1.217$, $p > 0.05$). Similarly, the negative soil visibility values are representative and expected for mature oil palm tree covered areas[32], as healthy vegetation absorbs visible light and reflects near-infrared light, leading to lower BSI index values (cf. Fig. 2a). Soil visibility (Sample 1: $\beta = -1.228$, $p < 0.01$; Sample 2: $\beta = -2.086$, $p < 0.01$) has a significant negative effect on oil palm plantation coverage in both samples, which is expected given the negative correlation between higher levels of plantation coverage and soil visibility. We find unimodal distributions for all environmental indices across all plantation types, with normally distributed values in the expected value ranges for the indices for agricultural land use of similar or comparable crops (cf. Fig. 2b), indicating no differences in environmental influences across plantation types. We also

controlled for self-produced and outsourced production, as indicated by the plantation type, to rule out alternative explanations pertaining to the ownership scheme under which the plantations are operated (cf. Fig. 3). Plantation output being self-produced (Sample 1: $\beta = 0.041$, $p < 0.01$; Sample 2: $\beta = 0.11$, $p > 0.05$) has a significant, positive effect only on the coverage of plantations certified in 2023, so five years post-announcement of the RSPO P&C 2018. Outsourced oil palm production (Sample 1: $\beta = -0.017$, $p > 0.05$; Sample 2: $\beta = -0.001$, $p > 0.05$) has no significant effect on the coverage of plantations. After controlling for the above-mentioned control variables, our findings remain robust and we observe a significant effect of certification obtainment on oil palm plantation coverage over time, before and after certification obtainment.

Our results indicate that obtainment of the RPSO certification resulted in decreased plantation efficiency, measured as the percentage of palm oil coverage on plantations both while they were preparing for certification and after certification. Plantations became less efficient at producing palm oil, although this is not the aim of the RSPO certification[33]. As the reduction in efficiency cannot be fully attributed to external demand dynamics (palm oil price) or environmental conditions (vegetation health and groundwater availability), the notable decreases in oil palm plantation efficiency before and after certification likely constitute an unintended consequence of certification practices. This conclusion holds given the persistence of our results even when accounting for statistically significant control variables. Figure 4 depicts a visualization of our analysis results for two exemplary plantations in our dataset. The outside supplier estate certified in 2018 under the RSB mill depicted in the upper image experienced a 21.97 percentage points total decline in oil palm coverage from 2018 until 2023. The coverage declined by 23.82 percentage points from 2018 to 2020 and increased by only 1.85 percentage points between 2020 and 2023. The own estate plantation certified in 2023 under the RSB mill depicted in the lower image was subject to a total decline of 18.74 percentage points in oil palm tree coverage from 2018 until 2023. The coverage declined by 8.22 percentage points from 2018 to 2020, and by 10.52 percentage points from 2020 until 2023.

Our results for Sample 1 (Plantations Certified in 2023), depicted in Model 2 in Table 2, indicate a decrease in plantation efficiency in most of the years leading up to the certification. The decrease, as compared to the reference year 2018, in which the RSPO P&C 2018 certification language was publicly announced, begins in 2019 ($\beta = -0.080$, $p < 0.001$) and intensifies in the years immediately before the certification: 2021 ($\beta = -0.195$, $p < 0.001$) and 2022 ($\beta = -0.252$, $p < 0.01$), except for 2020, when the effect, albeit negative, was non-significant. Our results for Sample 2 (Plantations Certified in 2018) indicate a long-term negative effect of certification obtainment. Compared to the reference year 2018, Sample 2, which obtained certification in 2018, shows a strong decline in coverage in

**Table 2 | The effects of RSPO certification on oil palm plantation efficiency**

| | Dependent variable: Oil palm plantation efficiency | | | |
| --- | --- | --- | --- | --- |
| | Sample 1: Plantations certified in 2023 | | Sample 2: Plantations certified in 2018 | |
| | (Model 1) | (Model 2) | (Model 3) | (Model 4) |
| Self-Produced | 0.041** | 0.041** | 0.011 | 0.011 |
| | (0.015) | (0.015) | (0.024) | (0.024) |
| Outsourced | −0.017 | −0.017 | −0.001 | −0.001 |
| | (0.013) | (0.013) | (0.018) | (0.018) |
| Vegetation health | 0.742* | 0.753* | 0.709 | 0.751 |
| | (0.306) | (0.307) | (0.589) | (0.565) |
| Groundwater availability | 2.336 | 2.349 | 1.107 | 1.217 |
| | (1.537) | (1.587) | (2.021) | (2.107) |
| Soil visibility | −1.234** | −1.228** | −2.147** | −2.086** |
| | (0.405) | (0.408) | (0.803) | (0.779) |
| Palm oil price | 0.00002*** | 0.0001*** | 0.00000 | 0.0001* |
| | (0.00001) | (0.00001) | (0.00001) | (0.00003) |
| Year 2017 | | 0.004 | | 0.165*** |
| | | (0.076) | | (0.027) |
| Year 2019 | | −0.080*** | | −0.175** |
| | | (0.020) | | (0.062) |
| Year 2020 | | −0.052 | | −0.079* |
| | | (0.027) | | (0.035) |
| Year 2021 | | −0.195*** | | −0.169** |
| | | (0.039) | | (0.055) |
| Year 2022 | | −0.252** | | −0.133 |
| | | (0.084) | | (0.104) |
| Constant | −3.101* | −3.207* | −2.459 | −2.639 |
| | (1.471) | (1.530) | (1.950) | (2.054) |
| Observations | 665 | 665 | 336 | 336 |
| Marginal $R^2$ | 0.285 | 0.379 | 0.326 | 0.486 |
| Conditional $R^2$ | 0.450 | 0.493 | 0.635 | 0.628 |
| Log likelihood | 319.191 | 312.927 | 180.706 | 175.559 |
| Akaike Inf. Crit. | −566.382 | −543.854 | −289.412 | −269.117 |
| Bayesian Inf. Crit. | −404.390 | −359.362 | −151.996 | −112.616 |

2018 is taken as reference level for "year". The category "small-grower plantation" is omitted from the plantation type.
*$p < 0.05$ **$p < 0.01$ ***$p < 0.001$ (two-tailed).

the three years directly following the certification: 2019 ($\beta = -0.175$, $p < 0.01$), 2020 ($\beta = -0.079$, $p < 0.05$), and 2021 ($\beta = -0.169$, $p < 0.01$).

We further explored whether plantation type influences oil palm plantation efficiency before and after the certification. A Kruskal–Wallis rank sum test[34] assessing the differences in coverage across the plantation types in our dataset revealed a significant effect of the plantation type on oil palm plantation coverage levels (Kruskal–Wallis chi-squared = 14.905, $df = 2$, $p < 0.001$). A Dunn's post-hoc test[35,36] indicated significantly higher coverages for own estate plantations than outside supplier estates ($z = 3.44$, $p < 0.001$) and small-grower plantations ($z = 3.59$, $p < 0.001$). No significant differences between the outside supplier plantation types (outside supplier estates and outside supplier small growers) were observed ($z = -0.28$, $p = 0.78$). Combining this result with plantation sizes by plantation type and corresponding certification years (cf. Fig. 3), it seems that CorpPalm's own estate plantations differ from outside supplier plantations both in preference for certification year and coverage levels: Own estates are being certified earlier and their coverage decreased less in this process compared to outside

**Table 3 | Environmental indices for the analyzed plantations over the study period**

| Year | Vegetation health (NDVI) | Soil visibility (BSI) | Groundwater availability (NDMI) |
| --- | --- | --- | --- |
| 2017 | 0.668 | −0.723 | 0.984 |
| 2018 | 0.712 | −0.761 | 0.988 |
| 2019 | 0.682 | −0.737 | 0.986 |
| 2020 | 0.675 | −0.733 | 0.984 |
| 2021 | 0.699 | −0.747 | 0.986 |
| 2022 | 0.705 | −0.756 | 0.988 |
| 2023 | 0.707 | −0.758 | 0.987 |

suppliers and small growers. Supplementary Table 1 presents further descriptive statistics.

## Discussion

By utilizing satellite data to study the impact of sustainability certifications on production in the Malaysian palm oil industry, we were able to overcome information asymmetries and data verification gaps by examining longitudinally, remotely, and at scale how the plantation efficiency of CorpPalm changed before and after RSPO certification obtainment. Obtaining this data on our dependent and control variables would not have been possible without the use of satellite data, as the producer does not publicly disclose data on the efficiency of their plantations, nor do they measure vegetation health, soil moisture, or soil coverage.

Despite the benefits, our analysis has some limitations. Our analysis is limited to plantations located in the Sabah region of Malaysia. While the region is a core location for oil palm plantations and comparable to other palm oil producing areas in Malaysian or Indonesian Borneo[37], local influencing factors may inhibit the generalizability of our findings to all oil palm plantations subject to RSPO certification globally. To minimize limitations from local factors, our analysis includes indices of external environmental conditions as control variables. Our results are therefore comparable to other oil palm plantations experiencing negligible changes or similar values in groundwater availability and vegetation health. Whether our findings may generalize to other crops and agricultural sectors is dependent on the influence of the criteria of the voluntary certification programs under investigation. Our main data source, satellite data, only permits measuring vegetation visible from space. While the spatial resolution of our data source is considered high and suitable for land use identification at 10 m, it could nonetheless introduce biases due to the misidentification of young or sparsely planted oil palm trees. The observed region of Sabah in Malaysia is also subject to severe and frequent cloud coverage stemming from higher cloud fractions in tropical regions[38]. We minimize the impact of such phenomena through the application of best practices for cloud coverage removal and up-sampling of our data. Even though the goal of our analysis is to investigate the impact of RSPO certification on plantation efficiency, we would like to point out that environmental variables and market prices are not the only, and perhaps, not the main factor that is responsible for changes in production efficiency. Other factors might include but are not limited to, government incentives and national policy shifts, seedling quality, fertilizer use, harvesting practices, pests not detected via the applied indices, further economic factors such as input costs, and climate change effects.

Our main finding that RSPO certification obtainment resulted in production decisions which decreased plantation efficiency both before and after certification contributes to the emerging body of studies investigating unintended consequences of market-based instruments of sustainability governance[4] and unacknowledged impacts of corporate social responsibility[3]. We deem the decrease in plantation production efficiency to be an unintended consequence of certification obtainment, as none of the RSPO certification language demands or describes as desirable a long-term decrease in plantation efficiency levels. As strategic effects of compliance

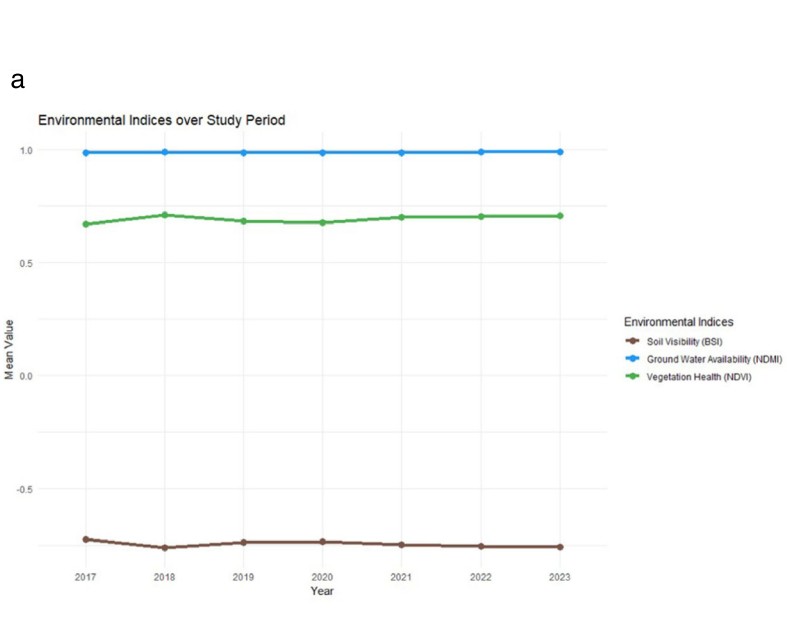

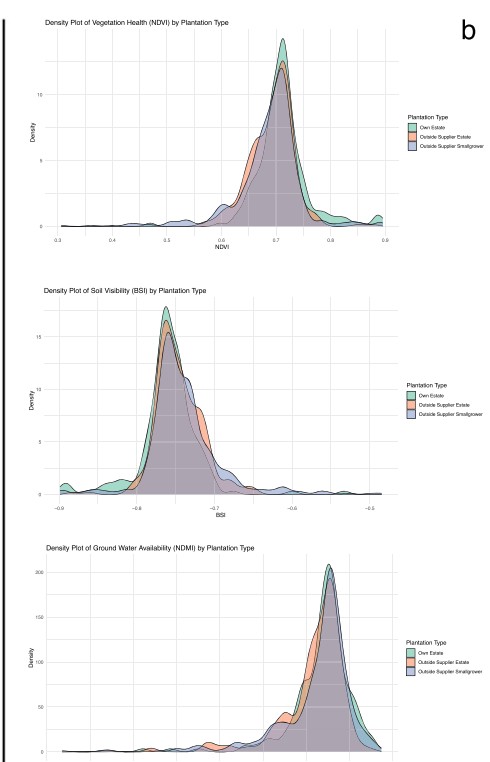

**Fig. 2 | Environmental indices of oil palm plantations. a** Means of environmental indices over time per year, with BSI indicated in brown, NDMI indicated in blue, and NDVI indicated in green. **b** Density plots of environmental indices by plantation types, with own estates represented in green, outside supplier estates represented in orange, and outside supplier small-grower plantations represented in purple. These figures were created using the R packages dplyr[56], tidyr[57], and ggplot2[58].

pertain over time and well beyond obtainment of the certification, up to three years post certification for Sample 2, the unintended consequences appear especially grave. In extreme cases, the continued decrease in plantation production efficiency may push oil palm producers to outsource their production to non-certified third-party suppliers to make up for the long-term loss in plantation efficiency. Such production expansions traceably occur outside the tropical rainforest biomes, which are not protected by zero-deforestation commitments but equally important for biodiversity[27]. Informal mills in particular are key drivers of rapid land use change and deforestation associated with palm oil production expansions[39].

During our seven-year investigation period, all of CorpPalm's plantations became certified. Furthermore, during that time, the plantations clearly made changes to their production strategies to guarantee compliance with the RSPO certification, even when the effect was negative in terms of production efficiency. Thus, our results encourage further exploration of market-based sustainability governance mechanisms and critical consideration of their consequences. Previous research has unveiled possible remedies and alternatives for agricultural expansion in the palm oil industry, including set-aside approaches[40], i.e., strategies for uncultivated agricultural landscape parts reducing environmental impacts without compromising productivity, tree islands[41], i.e., islands of native trees enriching oil palm landscape, mechanical weeding[42], support of independent smallholders rather than plantation corporations[43], or intensification on existing plantations[44]. Particularly the latter approach stands in stark contrast to the decrease in plantation efficiency and tree coverage unveiled by our study and therefore further emphasizes the need to reevaluate extant certification criteria to prevent this de-intensification on plantations and further deforestation and plantation expansion beyond the boundaries of the certification.

### Contributions and implications for the RSPO certification
Future iterations of the RSPO certificate should consider unintended effects of certification, particularly potential trade-offs from cropland expansion[45]

resulting from decreases in production efficiency on certified cropland. As of January 8, 2025, the 2024 RSPO P&C which was formally adopted on November 13, 2024 and shall become effective and binding on November 13, 2025 does not, to the best of our knowledge, explicitly contain such considerations[46]. The revised standards should be adapted to reference possible unintended consequences and explicitly highlight the potential of their occurrence among producers that have obtained or are planning to obtain certification. Particularly outsourced oil palm plantations, such as those of small-growers, should be informed of the negative effects of certification on their plantation's efficiency to avert unintended consequences before and after certification. Future review rounds should therefore facilitate dialogue concerning unintended consequences for production efficiency and plantation coverage and adapt existing criteria without sacrificing the current benefits of the certification. The Malaysian government has strongly encouraged sustainable practices in palm oil production, including RSPO certification. For policymakers our findings should therefore be taken into consideration particularly as certification standards become standard practices or officially supported and encouraged by the state.

## Methods
### Data selection
We opted for Sentinel-2 multispectral imagery data to analyze palm oil production with an unsupervised $k$-means algorithm. Sentinel-2 is particularly well suited for research on land cover and use, as it provides a higher spatial resolution than for example MODIS data and thus permits the distinction of objects otherwise too small for recognition, such as small-grower agricultural fields. Our data collection encompasses geolocated plantation data for one of the largest palm oil producers in Malaysia. From publicly available supplier and sourcing statistics, which include detailed maps of individual plantations, we identified geolocated shapefiles for 144 plantations of CorpPalm in Sabah, Malaysia, as shown in Fig. 5. RSPO and CorpPalm were contacted prior to publication. CorpPalm did not provide

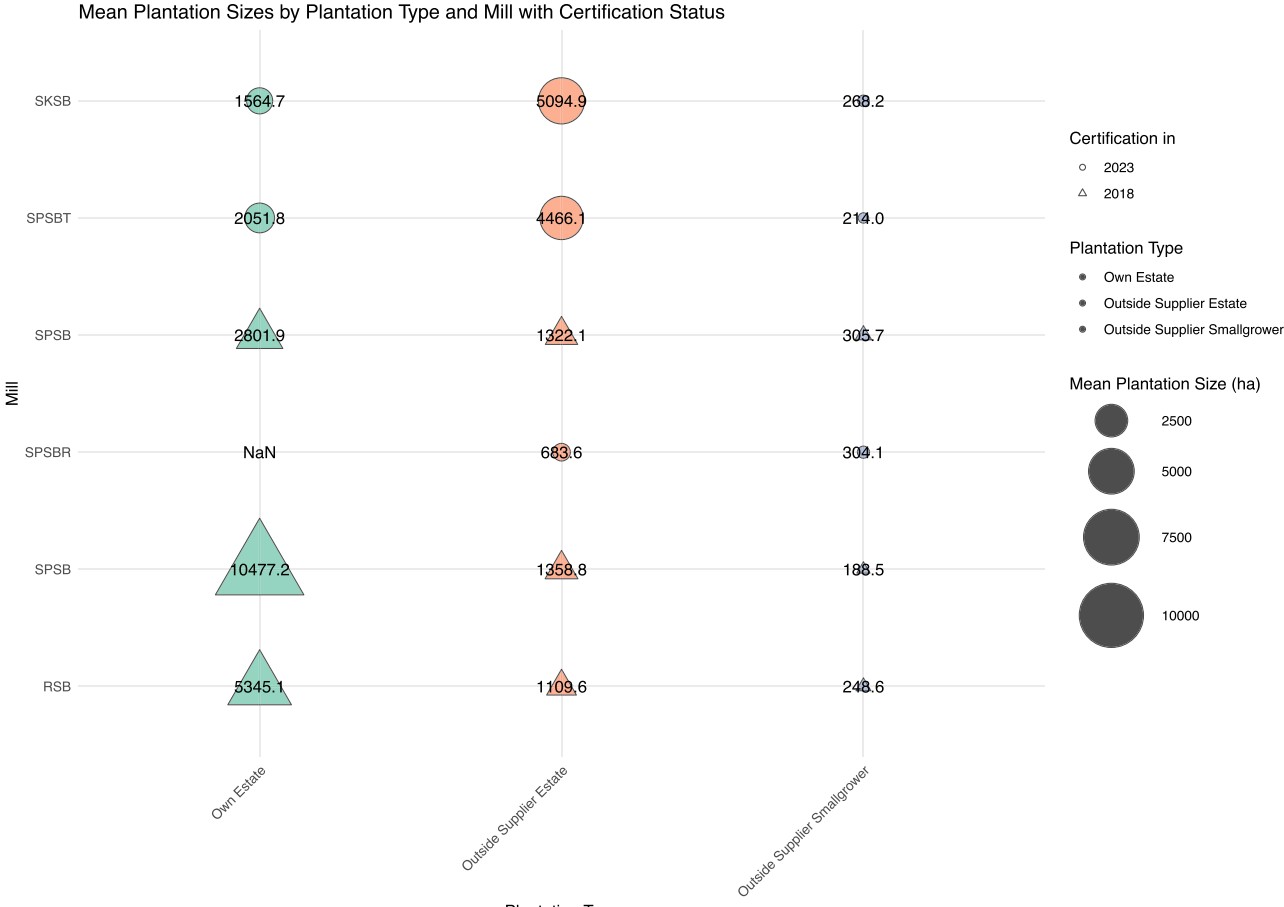

**Fig. 3 | Mean plantation sizes by plantation type and mill with corresponding certification years.** The values reported indicate the size of plantations that were certified in either 2018 (represented by triangular shapes) or 2023 (represented by circular shapes) under the respective mills in our dataset, categorized by plantation type. Mill names are indicated through acronyms, and the mean plantation sizes are indicated by the size of triangles and circles. Mills with comparatively more own estate plantation areas were certified in 2018, as compared to mills with comparatively more outside supplier estate, which were certified in 2023. This figure was created using the R packages dplyr[56], and ggplot2[58].

feedback. The RSPO provided feedback and was informed of the methodology and conclusions.

**Data collection and preparation**
To investigate our research problem on certification and oil palm plantation efficiency, we investigate how rates of mature oil palm tree coverage on CorpPalm's oil palm plantations changed pre- and post-certification. The change in tree coverage serves as a proxy for oil palm plantation efficiency through the density of trees on a given plantation area. We collected multispectral imagery tiles from the ESA Sentinel-2 (downloaded from the Copernicus Data Space Ecosystem) for Sabah, Malaysia, to measure longitudinal changes in palm oil tree growth patterns. Sentinel-2 is part of the ESA's Copernicus program and provides high-resolution multispectral imagery at 13 spectral bands to monitor among others land use, vegetation, and soil from the satellites Sentinel-2A and Sentinel-2B. Sabah has considerable cloud cover. We therefore first applied ESA-provided cloud coverage maps to remove cloud coverage. Sentinel-2 has a five-day revisit period at the equator thanks to its wide swath width at 290 km, which allowed us to collect enough data to create accurate cloud-free composites of the area. For each tile-year dyad, we made a pixel-by-pixel comparison of three separate multispectral imagery captures taken over a very short period of time to identify and remove additional cloud coverage (Delta Method)[47,48] in addition to the readily provided cloud coverage maps by ESA. The cloud-free tiles were then merged together to form cloudless tile composites.

This process was repeated for the entire period of our study. The final cloud-free tile composites comprised all 13 spectral bands and spanned four 100-km² regions. The RGB and Near-Infrared (NIR) bands of Sentinel-2 are provided at 10 m spatial resolution, the Red Edge and Short-Wave Infrared (SWIR) bands at 20 m, and the atmospheric correction bands at 60 m. In order to prepare our band composites, we up-sampled all of the bands to a 10 m resolution with a linear kernel[49]. This up-sampling ensured that all bands for our composites were consistent and at the same level of detail, simplifying data integration and analysis later on. Our final dataset encompasses a total of four Military Grid Reference System (MGRS) cloudless tile composites (50NNM, 50NPL, 50NNN, and 50NMM) of 100 km² at 10 m resolution, which thus equals about 60 Sentinel-2 captures. Our units of analysis are the individual plantations certified under CorpPalm in the Sabah region in Malaysia from our plantation shapefiles ($n = 144$, of which 48 were certified in 2018 and 96 in 2023).

**Measures**
Our dependent variable for analysis, oil palm tree coverage, i.e., the percentage of the plantation covered by palm trees, was obtained through land cover detection. We used an unsupervised $k$-means cluster analysis[50] to group similar spectral signals to extract data on the dominant features of different land cover types in our dataset. We first applied the $k$-means cluster analysis to a single tile capture aggregate[51]. The clusters were then classified

**Fig. 4 | Longitudinal visualization of changes in oil palm coverage of two exemplary plantations of CorpPalm (MGRS Tile 50 NNM).** The upper image shows Sentinel-2 cloud-free annual RGB composites from 2018, 2020, and 2023 of an outside supplier estate certified in 2018 under the RSB mill in our dataset (MGRS Tile 50NNM). The lower images show Sentinel-2 cloud-free annual RGB composites from 2018, 2020, and 2023 of an own estate plantation certified in 2023 under the SKSB mill in our dataset (MGRS Tile 50NNM). Pictures in the lower rows of both images virtually correspond to the upper pictures and visualize the results of the data analysis with an unsupervised $k$-means algorithm to identify land cover and use of the plantations and thereby identify the coverage of oil palm trees on the observed plantations. Oil palm coverage is depicted in yellow and includes mature and immature vegetation. The images are overlaid with the shapefile of the plantation in red.

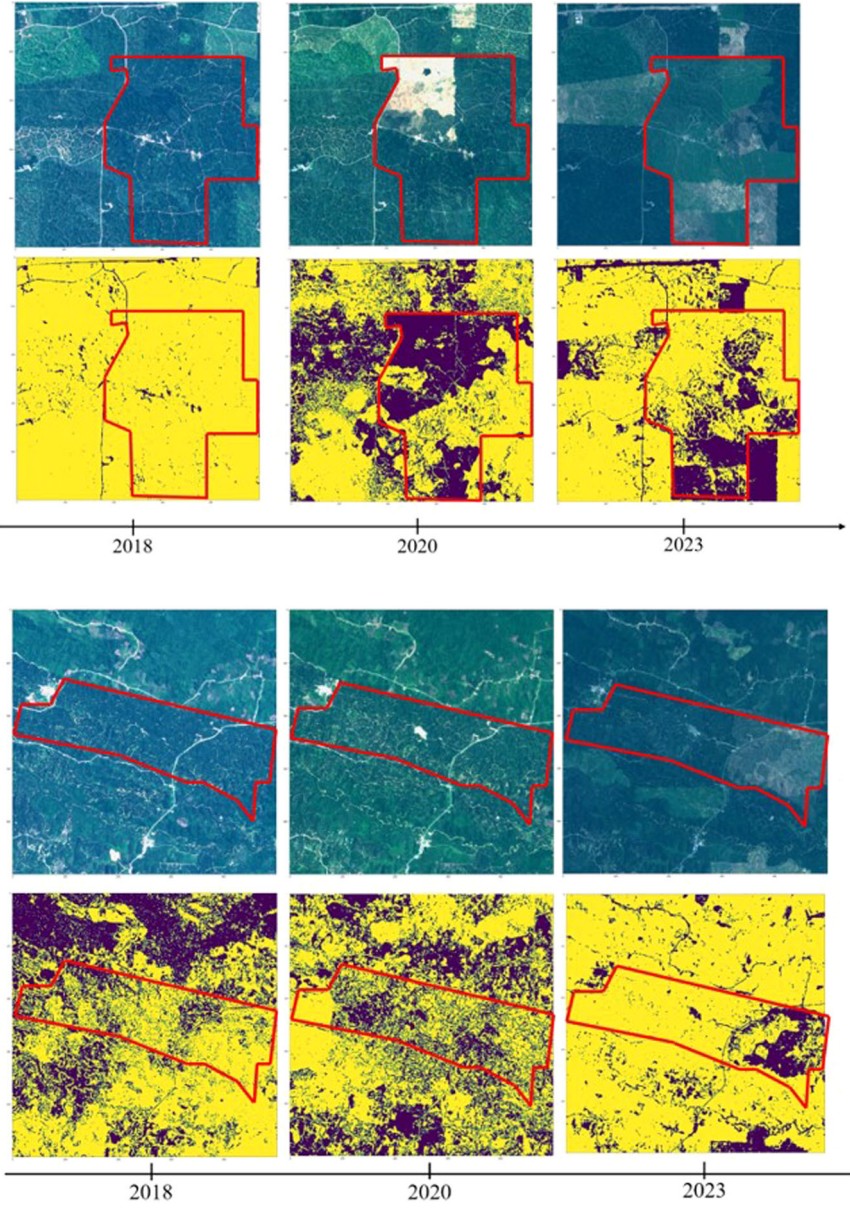

by hand to screen for mature oil palm trees, immature oil palm trees, other vegetation, and other artificial and natural artifacts. The cluster centers were saved and reapplied to all tile-year dyads. We extracted the cluster estimates with the plantation shape files and averaged them.

## Control variables

To control for potential confounding factors, we measure vegetation density, soil moisture, and soil coverage using band composites. We calculate three indices commonly employed for monitoring land cover and vegetation development. The indices are calculated based on the different bands in our multispectral satellite data. The NDVI, NDMI, and BSI are widely used to monitor vegetation health[52], soil moisture[53], and soil visibility[32].

The NDVI measures the difference between the near-infrared (B08) and red multispectral band (B04): $NDVI = \frac{B08-B04}{B08+B04}$. Healthy vegetation reflects more near-infrared and absorbs more red light. Higher NDVI values thus indicate healthier vegetation, whereas lower values indicate less or no vegetation in the observed area. The NDMI is sensitive to moisture levels in vegetation, which allows its application for e.g., water stress analysis or irrigation monitoring, and is calculated using the near-infrared (B08) and

shortwave infrared band (B11): $NDMI = \frac{B08-B11}{B08+B11}$. Higher values indicate lower levels of water content in the observed area, thus serving as a potential alternative explanation for lower vegetation density due to greater water stress. The BSI highlights barren ground, non-vegetated areas, and buildings or dwellings and is calculated using the blue (B02), red (N04), near-infrared (B08), and shortwave infrared (B11) band: $BSI = \frac{(B11+B04)-(B08+B02)}{(B11+B04)+(B08+B02)}$. The fidelity of all indices is contingent on the quality of the multispectral data and the specific characteristics of the study area. As the data utilized in this study is focused on clearly designated plantation areas, no unintended identification of e.g., buildings needs to be considered in BSI results. As with our tile aggregates, we first created band composite images for NDVI, NDMI, and BSI. We then extracted the spatial areas in our oil palm plantation shapefiles. Finally, the results for each of the indices were averaged across plantation-year dyads. We controlled for the type of oil palm plantation, differentiating between self-produced and outsourced production. Data on self-produced and outsourced production was obtained through CorpPalm's publicly available documents and included as a categorical control variable. Additionally, we controlled for the annual trade price of Malaysian palm oil[54].

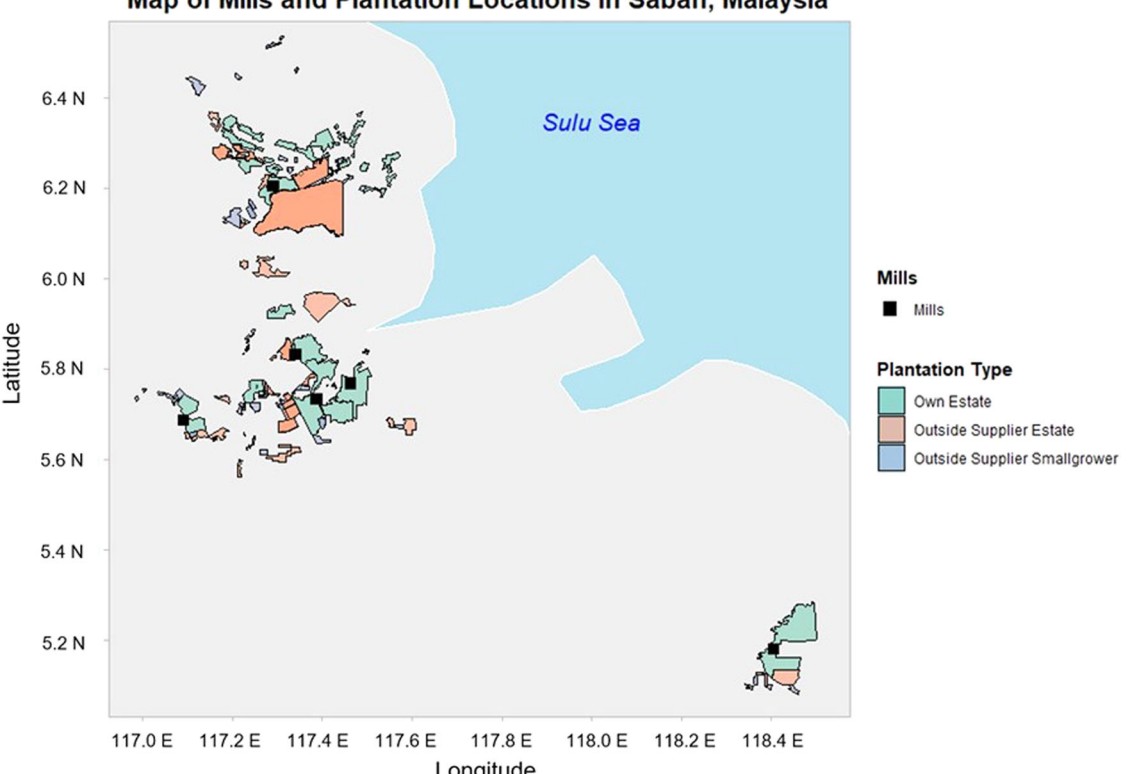

**Fig. 5 | Locations and types of analyzed mills and oil palm plantations in Sabah, Malaysia.** Excerpt of the mill locations and the shapefiles of exact plantation locations obtained from publicly available documents on the website of CorpPalm. This map was created using the following R packages: ggplot2[58], sf[59], dplyr[56], and maps[55]. Own estate plantations, i.e., plantations directly owned by CorpPalm, are depicted in green. Outside supplier estate plantations are depicted in orange. Small-grower plantations, i.e., plantations managed by CorpPalm under small-grower schemes, are depicted in purple. Intense colorization of shapefiles represents plantations noted as supplier plantations under two or more mills of CorpPalm certified by the RSPO.

## Data availability

The Sentinel-2 multispectral imagery tiles are publicly available from the ESA Sentinel-2 via the Copernicus Data Space Ecosystem: https://dataspace.copernicus.eu/explore-data/data-collections/sentinel-data/sentinel-2. The palm oil mill and plantation data were obtained from publicly available documents on the website of CorpPalm (pseudonym). Information on the RSPO certification status of mills and plantations certified under CorpPalm is publicly available from the RSPO certified growers search tool: https://rspo.org/search-members/certified-growers/ and also available from the authors upon request. The data on the price of Malaysian palm oil is available from Refinitiv Eikon[55]. The geotiffs and anonymized plantation data for the figures and tables are available for download from 10.5281/zenodo.14652527.

## Code availability

The composites were created using Julia, Cuda.jl, and Sentinel.jl. For the cluster analysis and empirical modeling, Python and R were used respectively. The R code is available from the authors upon reasonable request.

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

## Acknowledgements

We wish to thank the colleagues who provided feedback for their comments on the initial manuscript. We acknowledge financial support from the Swiss National Science Foundation (#207279 and #209466) and the University of St.Gallen through the GFF International Postdoc Fellowship grant (#1031596). We also thank the reviewers for their constructive comments, which helped to improve the manuscript.

## Author contributions

N.Z., M.H., and C.S. designed the study. M.H. collected and analyzed the satellite data. N.Z. collected the remaining data, and performed the modeling of the final dataset, supported by C.S. and G.G. N.Z. wrote the first draft of the manuscript. N.Z. and C.S. reviewed and edited the manuscript, supported by M.H. and G.G. N.Z., M.H., and C.S. acquired the applicable funding sources.

## Competing interests

The authors declare no competing interests.
