## [Transparent Peer Review file · Communications Earth & Environment]

Sustainable palm oil certification inadvertently affects production efficiency in Malaysia

Corresponding Author: Ms Nina Zachlod

Version 0:

Decision Letter:

Dear Ms Zachlod,

First of all, please let me apologise once again for the delay in sending a decision on your manuscript titled "Unintended Consequences of Roundtable on Sustainable Palm Oil (RSPO) Certifications in the Malaysian Palm Oil Industry". It has now been seen by 2 reviewers, and we include their comments at the end of this message. They find your work of interest, but some important points are raised. We are interested in the possibility of publishing your study in Communications Earth & Environment, but would like to consider your responses to these concerns and assess a revised manuscript before we make a final decision on publication.

We therefore invite you to revise and resubmit your manuscript, along with a point-by-point response that takes into account the points raised. Specifically, for publication in Communications Earth & Environment to be appropriate, we will need you to deepen your discussions, and in particular, to make a compelling case that sustainability certifications in line with the Roundtable on Sustainable Palm Oil (or preparations for certification) reduce palm oil plantation efficiency in Malaysia.

Please highlight all changes in the manuscript text file.

Please submit your point-by-point responses as a separate file, distinct from your cover letter where you can add responses to the Editors' comments that you do not want to be made available to the reviewers. Word files are preferred.

Important: The response to reviewers must not include any figures, tables or graphs. If you wish to respond to the reviewer reports with additional data in one of these formats, please add them to the main article or Supplementary Information, and refer to them in the rebuttal. Due to current technical limitations, any figures, tables, or graphs embedded in your rebuttal will not be included in the peer review file, if published.

Please use the following link to submit your revised manuscript, point-by-point response to the referees' comments (which should be in a separate document to any cover letter), a tracked-changes version of the manuscript (as a PDF file) and the completed checklist:

Link Redacted

We hope to receive your revised paper within six weeks; please let us know if you aren't able to submit it within this time so that we can discuss how best to proceed. If we don't hear from you, and the revision process takes significantly longer, we may close your file. In this event, we will still be happy to reconsider your paper at a later date, as long as nothing similar has been accepted for publication at Communications Earth & Environment or published elsewhere in the meantime.

Please do not hesitate to contact us if you have any questions or would like to discuss these revisions further. We look forward to seeing the revised manuscript and thank you for the opportunity to review your work.

Best regards,

Heike Langenberg, PhD
Chief Editor
Communications Earth & Environment

EDITORIAL POLICIES AND FORMATTING

Editorial Policy: [Policy requirements](https://www.nature.com/documents/nr-editorial-policy-checklist.pdf) (Download the link to your computer as a PDF.)

- Behavioural and social science
- Ecological, evolutionary & environmental sciences
- Life sciences

<https://www.nature.com/documents/nr-reporting-summary.zip>

Furthermore, please align your manuscript with our format requirements, which are summarized on the following checklist: [Communications Earth & Environment formatting checklist](https://www.nature.com/documents/commsj-phys-style-formatting-checklist-article.pdf)

and also in our style and formatting guide [Communications Earth & Environment formatting guide](https://www.nature.com/documents/commsj-phys-style-formatting-guide-accept.pdf) .

*** DATA: Communications Earth & Environment endorses the principles of the Enabling FAIR data project (<http://www.copdess.org/enabling-fair-data-project/>). We ask authors to make the data that support their conclusions available in permanent, publically accessible data repositories. (Please contact the editor if you are unable to make your data available).

All Communications Earth & Environment manuscripts must include a section titled "Data Availability" at the end of the Methods section or main text (if no Methods). More information on this policy, is available at <http://www.nature.com/authors/policies/data/data-availability-statements-data-citations.pdf>.

If a community resource is unavailable, data can be submitted to generalist repositories such as [figshare](https://figshare.com/) or [Dryad Digital Repository](http://datadryad.org/). Please provide a unique identifier for the data (for example a DOI or a permanent URL) in the data availability statement, if possible. If the repository does not provide identifiers, we encourage authors to supply the search terms that will return the data. For data that have been obtained from publically available sources, please provide a URL and the specific data product name in the data availability statement. Data with a DOI should be further cited in the methods reference section.

REVIEWER COMMENTS:

Reviewer #2 (Remarks to the Author):

As a researcher who have conducted a lot of research relating to oil palm, I found this manuscript to be interesting as it ventures into different angle which is yet to be explored that is assessing the unintended consequences (positive and negative) of RSPO certification on the Malaysian oil palm industry. The use of satellite data to analyse the impact of these certifications is innovative and offers valuable insights. However, there are several areas where the manuscript could be strengthened:

1. While the introduction provides a good overview of the RSPO certification and its importance, the literature review could be expanded to include more recent studies on the effectiveness and challenges of sustainability certifications in other agricultural sectors. This would provide a broader context and strengthen the argument. I also found the paper to on the bias side rather than carrying a neutral tone. For example, between line 67 to line 77, the authors are demonstrating their negative perspective on the oil palm industry by stating the industry is full of illegality and unethical reporting based on papers produced on the practice being conducted in a neighbouring country. Since the study is focusing on the oil palm scenario in Malaysia, it is inappropriate to make a general conclusion based on studies done in other countries.
2. Some of the results, particularly those related to the decrease in plantation efficiency, need further clarification. There are many influencing factors that could lead to this situation including the current environmental-related policy of the country, shift in the economy etc which are not being considered and or at least critically discussed. It would be beneficial to include more detailed explanations of the statistical models used and how the control variables were selected and interpreted.
3. The discussion section acknowledges some limitations, but a more detailed examination of potential biases and limitations of the satellite data, as well as the scope of the study, would add depth to the analysis. Additionally, discussing the generalizability of the findings to other regions and crops would be valuable.
4. The manuscript would benefit from a more detailed set of recommendations based on the findings. While the implications are discussed, specific suggestions for Malaysia's policymakers, industry stakeholders, and certification bodies would enhance the practical value of the research.

Specific Comments:

1. Abstract: The abstract is concise but could be more informative by briefly mentioning the key findings and their implications.
2. Figures and Tables: The figures and tables are useful, but the captions should be more descriptive. For instance, Figure 2's caption could explain what the satellite imagery reveals about palm oil tree coverage.
3. Conclusion: The conclusion should succinctly summarize the key findings and their broader implications. It currently overlaps with the discussion section and could be more focused.

Reviewer #3 (Remarks to the Author):

This manuscript focuses on the unintended consequences of the Roundtable on Sustainable Palm Oil (RSPO) certification on the efficiency of palm oil production in Malaysia. The main findings suggest that RSPO certification leads to a significant decline in plantation efficiency, both during the preparation phase and after certification. However, the manuscript needs improvement in several areas:

Results Section: The results need more detailed organization. The current analysis is too brief, and the results section is overly simplistic. For instance, a deeper analysis of the key factors causing the decline in palm oil production, such as whether it is due to changes in production strategies driven by RSPO certification, would be beneficial.

Language and Logic: The manuscript needs a thorough review to enhance accuracy in expression and improve logical flow.

Terminology: The term "palm oil plantation" is used extensively throughout the manuscript. It should be checked and possibly replaced with "oil palm plantation" to ensure correct usage. Please review and revise accordingly.

Specific Suggestions:

Line 80:

There is a repetition of "due to the." Please remove the redundant phrase.

Lines 91-93:

This sentence seems out of place here. Consider moving it to line 84 for better flow and coherence.

Lines 94-101:

The research findings should not be in the introduction. Consider relocating this content to a more appropriate section, such as the Results.

Results Section:

The current analysis is too simple. Consider providing more analysis on the key factors contributing to the decline in palm oil production, such as whether changes in production strategies due to RSPO certification are responsible for the decline.

Fig. 1-3:

The placement of these figures is inappropriate. Fig 1 belongs in the Methods section. Fig. 2 and Fig. 3 need to better represent the time differences (annual variations corresponding to the experimental results). These figures should also

include legends. Currently, they appear too simplistic, especially Fig. 3.

Line 191:

For the introduction of PalmCorp, please include a link to its website to allow readers to find more information. Add this link in an appropriate location.

Data Collection and Data Preparation:

For example, the phrase "The first step" in line 255 appears somewhat abrupt. The Data Collection and Data Preparation sections should be combined, as they belong together. Consider merging them.

Line 288:

Should the term "Control variables" be bolded?

Communications Earth & Environment is committed to improving transparency in authorship. As part of our efforts in this direction, we are now requesting that all authors identified as 'corresponding author' create and link their Open Researcher and Contributor Identifier (ORCID) with their account on the Manuscript Tracking System prior to acceptance. ORCID helps the scientific community achieve unambiguous attribution of all scholarly contributions. You can create and link your ORCID from the home page of the Manuscript Tracking System by clicking on 'Modify my Springer Nature account' and following the instructions in the link below. Please also inform all co-authors that they can add their ORCIDs to their accounts and that they must do so prior to acceptance.

If you experience problems in linking your ORCID, please contact the Platform Support Helpdesk.

Version 1:

Decision Letter:

Dear Ms Zachlod,

Your manuscript titled "Unintended Consequences of Roundtable on Sustainable Palm Oil (RSPO) Certifications in the Malaysian Palm Oil Industry" has now been seen by our reviewers. Please note that reviewer 2 is satisfied with revision, and comments from reviewer 3 are below. In light of their advice we are delighted to say that we are happy, in principle, to publish a suitably revised version in Communications Earth & Environment.

We therefore invite you to edit your manuscript to comply with our format requirements and to maximise the accessibility and therefore the impact of your work.

EDITORIAL REQUESTS:

****Please take care to match our formatting and policy requirements. We will check revised manuscript and return manuscripts that do not comply. Such requests will lead to delays. ****

SUBMISSION INFORMATION:

In order to accept your paper, we require the files listed at the end of the Editorial Requests Table; the list of required files is also available at <https://www.nature.com/documents/commsj-file-checklist.pdf> .

OPEN ACCESS:

Communications Earth & Environment is a fully open access journal. Articles are made freely accessible on publication. For further information about article processing charges, open access funding, and advice and support from Nature Research, please visit <https://www.nature.com/commsenv/open-access>

Link Redacted

Best regards,

Martina Grecequet, PhD
Senior Editor,
Communications Earth & Environment
@CommsEarth

REVIEWERS' COMMENTS:

Reviewer #3 (Remarks to the Author):

This manuscript has been improved. I have no further comments.

Response to Reviewer's Comments on Manuscript COMMSENV-24-1636-T

Title: Unintended Consequences of Roundtable on Sustainable Palm Oil (RSPO) Certifications in the Malaysian Palm Oil Industry

Overall Response: Thank you for the opportunity to revise and resubmit our paper. We appreciate the detailed feedback we received from the reviewers. The review process was a great help in improving the quality of our manuscript. In the revision, we have addressed all questions, suggestions and concerns raised. All changes to the manuscript text are highlighted in yellow in the manuscript. Following, we provide detailed responses to the reviewer comments. We present reviewers' comments in bold and our response in regular typeface.

REVIEWER 2

As a researcher who have conducted a lot of research relating to oil palm, I found this manuscript to be interesting as it ventures into different angle which is yet to be explored that is assessing the unintended consequences (positive and negative) of RSPO certification on the Malaysian oil palm industry. The use of satellite data to analyse the impact of these certifications is innovative and offers valuable insights. However, there are several areas where the manuscript could be strengthened:

Response: We very much appreciate your positive words about our paper. We also appreciate your constructive comments and recommendations. We address each of your points below and point to corresponding changes in the manuscript.

1. While the introduction provides a good overview of the RSPO certification and its importance, the literature review could be expanded to include more recent studies on the effectiveness and challenges of sustainability certifications in other agricultural sectors. This would provide a broader context and strengthen the argument. I also found the paper to on the bias side rather than carrying a neutral tone. For example, between line 67 to line 77, the authors are demonstrating their negative perspective on the oil palm industry by stating the industry is full of illegality and unethical reporting based on papers produced on the practice being conducted in a neighbouring country. Since the study is focusing on the oil palm scenario in Malaysia, it is inappropriate to make a general conclusion based on studies done in other countries.

Response: Thank you for your positive evaluation of our introduction section. We now clearly delineate the boundaries of previous findings on the palm oil industry and explicitly state the country-context of previous studies. We have correspondingly adjusted the section you are referring to, to refrain from conveying a biased perspective on the industry as a whole and to prevent generalizations about the industry in Malaysia based on studies conducted in other countries. We have also expanded the literature review to include a paragraph on recent studies investigating the effectiveness of voluntary certification programs in other agricultural sectors (p. 3):

“Recent studies have focused predominantly on the socioeconomic¹ and financial² impacts of the RSPO. The primary reason for the lack of studies on the environmental consequences of palm oil sustainability certificates and RSPO certification specifically is the demanding nature of data

¹ Santika, T. et al. Impact of palm oil sustainability certification on village well-being and poverty in Indonesia. *Nat. Sustain.* **4**, 109–119 (2021).

² Tey, Y. S., Brindal, M., Djama, M., Hadi, A. H. I. A. & Darham, S. A review of the financial costs and benefits of the Roundtable on Sustainable Palm Oil certification: Implications for future research. *Sustain. Prod. Consum.* **26**, 824–837 (2021).

collection in this study context. The few studies done have concentrated on Indonesia, where the palm oil industry has been associated with issues of legality and self-reporting practices³. Previous research on the effect of the RSPO on people and the environment has unveiled heterogeneous effects of the certification on deforestation and forest protection in Indonesia⁴. Other studies have investigated comparable certificates and voluntary certification programs in other agricultural sectors. A study of the Forest Stewardship Council (FSC) in Kalimantan, Indonesia⁵, has found a positive environmental impact of the certification program, reducing aggregate deforestation by five percentage points between 2000 and 2008. Whilst it had no statistically significant impact on e.g. fire incidences, it reduced the incidences of air pollution by 31% in the observed time period. A study on the reduced emissions from deforestation and forest degradation (REDD+) programs across the Global South⁶ has found positively significant yet moderately sized average environmental impacts and welfare-neutral to slightly positive socioeconomic impacts. Research on other (non-RSPO) natural resource policies designed to curtail production's environmental impact however found that they may result in unintended consequences, such as the displacement of environmental impacts⁷."

2. Some of the results, particularly those related to the decrease in plantation efficiency, need further clarification. There are many influencing factors that could lead to this situation including the current environmental-related policy of the country, shift in the economy etc which are not being considered and or at least critically discussed. It would be beneficial to include more detailed explanations of the statistical models used and how the control variables were selected and interpreted.

Response: Thank you for this comment, which we paid much attention to and made many changes accordingly. We now present our control variables (controlling for the influence of external demand dynamics or environmental conditions) in more detail and provide further analysis on pages 6-10. We detail that our findings remain robust and we observe a significant effect of certification obtainment on oil palm plantation coverage over time after controlling for the control variables as alternative explanatory factors for the decline. In addition, we have added Figures 1-3, which provide an in-depth analysis of the plantation types, their certification obtainment and area (Figure 1), the environmental indices we use to control for vegetation health, soil visibility, and groundwater availability (Figure 2), and a comprehensive overview of the mean plantation sizes by plantation type, mill and certification year (Figure 3), to our results section. Furthermore, we have adapted Figure 4 (previously Figure 2 and 3) to be a longitudinal visualization of the changes in oil palm coverage of two exemplary plantations from our dataset. It includes true-color composites of the plantations from 2018, 2020, and 2023 along with corresponding visualizations of our data analysis, which display the changes in oil palm tree coverage in an easily interpretable color scheme. We also address alternative influencing factors that could lead to the decline in plantation efficiency in our discussion section on page 16 (for the exact wording, please kindly refer to our answer to your point #3).

³ Astuti, R. et al. Making illegality visible: The governance dilemmas created by visualising illegal palm oil plantations in Central Kalimantan, Indonesia. *Land Use Policy* **114**, 105942 (2022).

⁴ Lee, J. S. H. et al. Does oil palm certification create trade-offs between environment and development in Indonesia? *Environmental Research Letters*, **15**, 124064 (2020).

⁵ Miteva, D. A., Loucks, C. J. & Pattanayak, S. K. Social and Environmental Impacts of Forest Management Certification in Indonesia. *PloS One* **10**(7): e0129675 (2015).

⁶ Wunder, S., Schulz, D., Montoya-Zumaeta, J.G. et al. Modest forest and welfare gains from initiatives for reduced emissions from deforestation and forest degradation. *Commun Earth Environ* **5**, 394 (2024).

⁷ Lewison, R. L., Johnson, A. F., Gan, J., Pelc, R., Westfall, K. & Helvey, M. Accounting for unintended consequences of resource policy: Connecting research that addresses displacement of environmental impacts. *Conserv. Lett.* **12**(3), e12628 (2019).

3. The discussion section acknowledges some limitations, but a more detailed examination of potential biases and limitations of the satellite data, as well as the scope of the study, would add depth to the analysis. Additionally, discussing the generalizability of the findings to other regions and crops would be valuable.

Response: Thank you for highlighting this possibility for improving our discussion section. Following your recommendation, we have extended our discussion of potential biases and limitations as well as the scope of the study, to include detailed examinations of the inherent limitations and their consequences for the generalizability of our findings (p. 15-16). The discussion of limitations now reads as follows (p. 15-16):

“Despite the benefits, our analysis has some limitations. Our analysis is limited to plantations located in the Sabah region of Malaysia. While the region is a core location for oil palm plantations and comparable to other palm oil producing areas in Malaysian or Indonesian Borneo⁸, local influencing factors may inhibit the generalizability of our findings to all oil palm plantations subject to RSPO certification globally. To minimize limitations from local factors, our analysis includes indices of external environmental conditions as control variables. Our results are therefore comparable to other oil palm plantations experiencing non-significant changes or similar values in groundwater availability and vegetation health. Whether our findings may generalize to other crops and agricultural sectors is dependent on the influence of the criteria of the voluntary certification programs under investigation. Our main data source, satellite data, only permits measuring vegetation visible from space. While the spatial resolution of our data source is considered high and suitable for land use identification at 10 m, it could nonetheless introduce biases due to the misidentification of young or sparsely planted oil palm trees. The observed region of Sabah in Malaysia is also subject to severe and frequent cloud coverage stemming from higher cloud fractions in tropical regions⁹. We minimize the impact of such phenomena through the application of best practices for cloud coverage removal and up-sampling of our data. Even though the goal of our analysis is to investigate the impact of RSPO certification on plantation efficiency, we would like to point out that environmental variables and market prices are not the only, and perhaps, not the main factor that is responsible for changes in production efficiency. Other factors might include but are not limited to, government incentives and national policy shifts, seedling quality, fertilizer use, harvesting practices, pests not detected via the applied indices, further economic factors such as input costs, and climate change effects.”

4. The manuscript would benefit from a more detailed set of recommendations based on the findings. While the implications are discussed, specific suggestions for Malaysia’s policymakers, industry stakeholders, and certification bodies would enhance the practical value of the research.

Response: Thank you for pointing out this possible improvement to our manuscript. Following your input we have added a separate sub-chapter titled “*Contributions and Implications for the RSPO Certification*” to our discussion. There, we provide more detailed recommendations focusing on the certification body and implications for them, as this is what we may derive implications for based on our findings on pages 17-18. We hope that this elucidation helps our readers better understand what can be done specifically to remedy the unintended consequences of the certificate unveiled by our findings. The inserted paragraph reads as follows:

“Contributions and Implications for RSPO Certification. Future iterations of the RSPO certificate should consider unintended effects of certification, particularly potential trade-offs from cropland

⁸ Gaveau, D., Sheil, D., Husnayaen et al. Rapid conversions and avoided deforestation: examining four decades of industrial plantation expansion in Borneo. *Sci Rep* **6**, 32017 (2016).

⁹ Xu, R., Li, Y., Teuling, A.J. et al. Contrasting impacts of forests on cloud cover based on satellite observations. *Nat Commun* **13**, 670 (2022).

expansion¹⁰ resulting from decreases in production efficiency on certified cropland. Currently, the Standards Review 2022-2024 draft that is the basis for the new RSPO Principles and Criteria 2024 does not contain such considerations¹¹. Revised standards should be adapted to reference possible unintended consequences and explicitly highlight the potential of their occurrence among producers that have obtained or are planning to obtain certification. Particularly outsourced oil palm plantations, such as those of small-growers, should be informed of the negative effects of certification on their plantation's efficiency to avert unintended consequences before and after certification. The ongoing review round should therefore facilitate dialogue concerning unintended consequences for production efficiency and plantation coverage and adapt existing criteria without sacrificing the current benefits of the certification. The Malaysian government has strongly encouraged sustainable practices in palm oil production, including RSPO certification. For policymakers our findings should therefore be taken into consideration particularly as certification standards become standard practices or officially supported and encouraged by the state. ”

Specific Comments:

1. Abstract: The abstract is concise but could be more informative by briefly mentioning the key findings and their implications.

Response: We have modified the abstract to explicitly state the key finding of our research and its core implication (p. 1):

“Our results indicate that, as they prepare for the certification and after obtaining it, palm oil producers engage in farming practices that decrease palm oil production efficiency. This unintended consequence, which is not an explicit goal of the certification and is ill suited to addressing the challenges of sustainable production, should be critically considered in future iterations of the RSPO certificate.”

2. Figures and Tables: The figures and tables are useful, but the captions should be more descriptive. For instance, Figure 2's caption could explain what the satellite imagery reveals about palm oil tree coverage.

Response: Following your recommendation, we have adjusted the captions for all figures accordingly. We now provide detailed explanations of how the satellite imagery permits the identification of land cover and use for the oil palm plantations under observation for Figure 4 (previously Figures 2 and 3) on page 14. The caption now reads as follows:

“Fig. 4. Longitudinal Visualization of Changes in Oil Palm Coverage of two Exemplary Plantations of CorpPalm (MGRS Tile 50 NNM). The upper image shows Sentinel-2 cloud-free annual RGB composites from 2018, 2020, and 2023 of an outside supplier estate certified in 2018 under the RSB mill in our dataset (MGRS Tile 50NNM). The lower images show Sentinel-2 cloud-free annual RGB composites from 2018, 2020, and 2023 of an own estate plantation certified in 2023 under the RSB mill in our dataset (MGRS Tile 50NNM). Pictures in the lower rows of both images virtually correspond to the upper pictures and visualize the results of the data analysis with an unsupervised k-means algorithm to identify land cover and use of the plantations and thereby identify the coverage of oil palm trees on the observed plantations. Oil palm coverage is depicted in yellow and includes mature and immature vegetation. The images are overlaid with the shapefile of the plantation in red.”

3. Conclusion: The conclusion should succinctly summarize the key findings and their

¹⁰ Schneider, J. M. et al. Effects of profit-driven cropland expansion and conservation policies. *Nat. Sustain* (2024).

¹¹ Roundtable on Sustainable Palm Oil. RSPO standards PC revision 2.0 draft. *Roundtable on Sustainable Palm Oil* (2024). <https://rspo.org/wp-content/uploads/RSPO-Standards-PC-Revision-2.0-Draft.pdf>

broader implications. It currently overlaps with the discussion section and could be more focused.

Response: Following the example of previous publications in *Communications Earth & Environment*, our previous submission did not include a separate conclusion section. We however see merit in your proposition and have therefore followed previous publications and separated our contributions¹² and implications¹³ from our discussion section through the inclusion of a separate sub-chapter titled “*Contributions and Implications for the RSPO Certification*” (p. 17-18). We have provided the content of this sub-chapter in our response to your main comment #4.

REVIEWER 3

This manuscript focuses on the unintended consequences of the Roundtable on Sustainable Palm Oil (RSPO) certification on the efficiency of palm oil production in Malaysia. The main findings suggest that RSPO certification leads to a significant decline in plantation efficiency, both during the preparation phase and after certification. However, the manuscript needs improvement in several areas:

Response: Thank you for your assessment of the manuscript. We are grateful for your constructive comments and recommendations, as they have helped us to improve the manuscript substantially. We address your points below and point to the corresponding changes in the manuscript.

Results Section: The results need more detailed organization. The current analysis is too brief, and the results section is overly simplistic. For instance, a deeper analysis of the key factors causing the decline in palm oil production, such as whether it is due to changes in production strategies driven by RSPO certification, would be beneficial.

Response: Per your suggestions, we have conducted more in-depth analysis of our data and the key factors potentially causing a decline in oil palm plantation coverage. We provide details of these additions in the response to your Point “Results Section” under “Specific Suggestions”.

Language and Logic: The manuscript needs a thorough review to enhance accuracy in expression and improve logical flow.

Response: We have revised the manuscript and improved the accuracy by e.g., including contextual information and thoroughly reviewing prior results. The revised version of the manuscript has also undergone a thorough review by a professional editor.

Terminology: The term “palm oil plantation” is used extensively throughout the manuscript. It should be checked and possibly replaced with “oil palm plantation” to ensure correct usage. Please review and revise accordingly.

¹² Tang, L. & Werner, T. T. Global mining footprint mapped from high-resolution satellite imagery. *Communications Earth & Environment*, **4**, 134 (2023).

¹³ Ordway, E. M. et al. Mapping tropical forest functional variation at satellite remote sensing resolutions depends on key traits. *Communications Earth & Environment*, **3**, 347 (2022).

Response: Benchmarking previous publications¹⁴, we have ensured that all mentions of the term that refer to oil palm trees or plantations have been reworded to “oil palm plantation” instead of “palm oil plantation”. We have retained the term ‘palm oil’ when referring to the product or the industry¹⁵.

Specific Suggestions:

Line 80: There is a repetition of “due to the.” Please remove the redundant phrase.

Response: Thank you for your careful read of our submission, we have removed the redundant phrase and thoroughly checked the manuscript to ensure correct writing throughout.

Lines 91-93: This sentence seems out of place here. Consider moving it to line 84 for better flow and coherence.

Response: We have moved the sentence to p. 4. We have also thoroughly reviewed and reorganized the introduction and results sections to improve flow and consistency.

Lines 94-101: The research findings should not be in the introduction. Consider relocating this content to a more appropriate section, such as the Results.

Response: We have relocated the indicated section to the results section on p. 11. We have also modified the abstract to explicitly state the key finding of our research and its core implication (p. 1), following the corresponding comment from Reviewer 2.

Results Section: The current analysis is too simple. Consider providing more analysis on the key factors contributing to the decline in palm oil production, such as whether changes in production strategies due to RSPO certification are responsible for the decline.

Response: Thank you for this comment, which is extremely valuable and important and which we have addressed in multiple ways throughout our manuscript. We have added Figures 1-3, which provide an in-depth analysis of the plantation types, their certification obtainment, and area (Figure 1), the environmental indices we use to control for vegetation health, soil visibility, and groundwater availability (Figure 2), and a comprehensive overview of the mean plantation sizes by plantation type, mill and certification year (Figure 3), to our results section. We have also adapted Figure 4 (previously Figure 2 and 3) to be a longitudinal visualization of the changes in oil palm coverage of two exemplary plantations from our dataset. It includes true-color composites of the plantations from 2018, 2020, and 2023 along with corresponding visualizations of our data analysis, which display the changes in oil palm tree coverage in an easily interpretable color scheme (cf. our response to your comment on Figures 1-3). We also present our control variables in more detail and provide further analysis on pages 6-10. We detail that our findings remain robust and we observe a significant effect of certification preparation and obtainment on oil palm plantation coverage over time after controlling for the control variables as alternative explanatory factors for the decline. We have also conducted a Kruskal-Wallis rank sum test and a Dunn’s post-hoc test to further analyze the possible influence of the plantation type on the decline in oil palm plantation coverage. In addition, we have significantly expanded our limitations section and explicitly state the limitations of our research and the measures we have taken to address them. Whilst we have undertaken significant effort to control for various

¹⁴ Srinivasan, U., Velho, N., Lee, J.S.H. et al. Oil palm cultivation can be expanded while sparing biodiversity in India. *Nat Food* 2, 442–447 (2021).

¹⁵ Santika, T., Wilson, K.A., Law, E.A. et al. Impact of palm oil sustainability certification on village well-being and poverty in Indonesia. *Nat Sustain* 4, 109–119 (2021).

influencing factors, we acknowledge that our study may not rule out every possible alternative explanation. We therefore also highlight that other factors that might potentially influence production efficiency prevail, by adding the following excerpt on page 16:

“Even though the goal of our analysis is to investigate the impact of RSPO certification on plantation efficiency, we would like to point out that environmental variables and market prices are not the only, and perhaps, not the main factor that is responsible for changes in production efficiency. Other factors might include but are not limited to, government incentives and national policy shifts, seedling quality, fertilizer use, harvesting practices, pests not detected via the applied indices, further economic factors such as input costs, and climate change effects.”

Fig. 1-3: The placement of these figures is inappropriate. Fig 1 belongs in the Methods section. Fig. 2 and Fig. 3 need to better represent the time differences (annual variations corresponding to the experimental results). These figures should also include legends. Currently, they appear too simplistic, especially Fig. 3.

Response: We have relocated Figure 5 (previously Figure 1) on page 16 to the Methods section and adapted the content to better represent the type of oil palm plantations analyzed, along with a more detailed legend. Figure 4 (previously Figures 2 and 3) on pages 13-14 now presents longitudinal depictions and visualization of two exemplary plantations and the application of our analysis in the form of annual variations for the duration of our study. Where applicable, we included legends or detailed descriptions of the figures in the caption.

Line 191: For the introduction of PalmCorp, please include a link to its website to allow readers to find more information. Add this link in an appropriate location.

Response: The goal of the study was to share how RSPO certification preparation and obtainment could have unintended production efficiency consequences. We picked one of the largest companies in Malaysia and South East Asia which operates palm oil plantations. The study was conducted using secondary data, specifically archival satellite imagery. We did not seek or secure permission from the company to name the corporate operating entity. Because the company is publicly traded, and this might be material information, we were advised to not disclose the name of the company. However, should curious individuals want to identify the company, it is easily identifiable by satellite data and land records given that the company owns large tracts of land in that area. Further identifying documents are available upon request from the authors, as indicated in the Data Availability section. To avoid confusion with existing companies, we have adapted the pseudonym to “CorpPalm” throughout the manuscript and highlighted clearly that the name is a pseudonym on p. 6. We have also provided the name of the company and a link to their website to the editor via the online submission system section “Manuscript Comment”. For review purposes, the name may be obtained from the editor.

Data Collection and Data Preparation: For example, the phrase “The first step” in line 255 appears somewhat abrupt. The Data Collection and Data Preparation sections should be combined, as they belong together. Consider merging them.

Response: Thank you for highlighting this need for further improvement of the structure of our manuscript. We have merged the sections and improved upon the phrasing (p. 19-20):

Line 288: Should the term “Control variables” be bolded?

Response: We have adjusted the subheading accordingly and bolded it.